# Investigation into the Rheological Properties and Microstructure of Silt/Crumb Rubber Compound-Modified Asphalt

**DOI:** 10.3390/polym15112474

**Published:** 2023-05-27

**Authors:** Lu Huang, Jiuguang Geng, Mingyuan Chen, Yanhui Niu, Wenhao Wang, Zichen Gao

**Affiliations:** School of Materials Science and Engineering, Chang’an University, Xi’an 710018, China; hl344024159@163.com (L.H.);

**Keywords:** sludge solidification, rubber asphalt, mechanical properties, modification mechanism

## Abstract

Near the coast of China, a large amount of sediment is produced during construction work. In order to mitigate the environmental damage caused by sediment and enhance the performance of rubber-modified asphalt effectively, solidified silt material and waste rubber were prepared to modify asphalt, and its macroscopic properties, such as viscosity and chemical composition, were determined via a routine physical test, DSR, Fourier Transform Infrared Spectroscopy (FTIR), and Fluorescence Microscopy (FM). The results show that, with the increase in powder particles and the addition of a certain amount of hardened mud, the mixing and compaction temperature of modified asphalt can be significantly increased—still reaching the design standard. In addition, the high thermal stability and fatigue resistance of the modified asphalt were clearly better than those of the ordinary asphalt. From the FTIR analysis, rubber particles and hardened silt only exhibited mechanical agitation with the asphalt. Considering that excessive silt might result in the aggregation of matrix asphalt, the addition of an appropriate amount of hardened solidified silt material can eliminate the aggregation. Therefore, the performance of modified asphalt was optimum when solidified silt was added. Our research can provide an effective theoretical basis and reference values for the practical application of compound-modified asphalt. Therefore, 6%HCS(6:4)-CRMA have better performance. Compared to ordinary rubber-modified asphalt, the composite-modified asphalt binder has better physical properties and a more suitable construction temperature. The composite-modified asphalt uses discarded rubber and silt as raw materials, which can effectively protect the environment. Meanwhile, the modified asphalt has excellent rheological properties and fatigue resistance.

## 1. Introduction

For the past few years, with the continuous augment of infrastructure construction in the coastal areas across China, more and more silt has accumulated annually, which can easily lead to a series of problems such as environmental pollution and urban space occupation. Additionally, there are more and more used car tires due to the gradual increase in the use of cars across the country. The treatment of asphalt by waste rubber powder is a potentially beneficial use of waste rubber [1]. When compared to base asphalt, rubber-modified asphalt (CRMA) can significantly strengthen pavement fatigue resistance, low temperature cracking resistance, and durability. CRMA can contribute to realizing excellent road performance and improving the service life of vehicles, thus reducing the consumption of energy and fuel [2]. Silty soil is characterized by high water content, high porosity ratio, and low strength or permeability, which is compressible. However, it is hard to achieve optimum efficiency in the physical blending of silt and asphalt. As a result, one of the crucial methods for improving silt characteristics is to solidify the silt [3].

Cement is considered to be the most commonly used curing agent because it is capable of rapid hydrate reaction with water in the sludge [4]. Tremblay et al. [5] studied the solidification of silty soil, finding that the better the solidified effect of the muck, the stronger and less ductile the solidified soil. Moreover, the compressive property is affected by the degree of solidification regarding the silty soil. S.N. Omkaret et al. [6] proposed that cement-solidified soft soil can reach an elevated curing strength. By establishing a simplified viscoelastic-continuity failure (S-VECD) theory based on viscous-continuity failure, the effects of different amounts and ages of cementitious materials on the properties of cementitious materials are investigated, so as to provide a theoretical basis for the popularization of cementitious materials. Chen et al. [7] believed that, with an increase in the amount of concrete, the hardening effect of the silt will improve. According to Ma et al. [8], the hydrated magnesium of the third phase, the fifth phase, and other hydrated components in CS forms a crystal network in the hydrated hardening products, thus improving the strength of CS. Qiu et al. [9] stated that the addition of cement can diminish the pore diameter of silty hardened soil, thus reducing its shrinkage. Chen et al. [10] conducted an exploration of the curing effect of different types of Portland concrete. The results indicated that conventional Portland concrete leads to the optimal curing effect. In addition, despite the inability to increase its mechanical properties, its good filling and remarkable impermeability can efficiently decrease the occurrence of water damage. Ray et al. [11] argued that CSH and CAH are the most important gelation products in hardened DS. From SEM and XRD analysis, it can be proved that the addition of NS promotes the formation of CSH in CDS system. Therefore, adding an appropriate amount of cement to solidify silt can be effective in lessening the porosity of silt. In summary, cement-solidified silt can significantly reduce its porosity and form a dense network structure.

Rubber powder is widely used in modified asphalt [12]. A new type of improved rubber adhesive prepared from rubber powder and petroleum additive can improve the rheological properties of petroleum products. According to Satherin Poovaneshvaran et al. [13], asphalt material modified by rubber fragments can exhibit excellent properties in terms of adhesion strength. Studies by Liu et al. [14] showed that the relative molecular weight, dissolution parameters, chemical structure, and thermal stability of rubber powder are different compared to matrix tar, making it a thermodynamically incompatible system. However, given the process compatibility presented by the two, physical mixing can be realized by mechanical mixing. Hu K et al. [15] found that CWTB can greatly intensify the heat resistance of base asphalt, which has a great influence on base asphalt. Liang et al. [16] examined that adding waste tire rubber to asphalt can significantly improve the viscosity by 60 degrees Celsius and enhance the ability to resist rutting or lasting deformation. Additionally, using a microscope, it was observed that there were almost no fine particles in the matrix, indicating an improved blending of the two materials. Ren SS et al. [17] found the CR swelling-degradation degree significantly affected the physical and viscoelastic properties of CR/SBS-modified high-viscosity binders. Gong et al. [18] discussed whether the addition of rubber powder and admixture to base asphalt can increase the elastic composition and invigorate its permanent deformation resistance. The research of D Singh et al. [19] revealed that CR plays a significant role in improving the adhesion and water corrosion resistance of CRMA cementitious materials. By means of DSR test, Zhang et al. [20] found that adding rubber powder to the flat asphalt can enhance the asphalt matrix’s high thermal resistance. Liang M. et al. [21] found that RPE in the composite additive plays a key role in improving the resistance to permanent deformation and reducing viscoplastic behavior. Rubber powder can usually greatly improve the high- and low- temperature properties of asphalt mixture [22,23]. Furthermore, since simple rubber materials have been unable to adapt to the current higher temperature environment, it is necessary to carry on in-depth research in order to heighten its high thermal resistance and water damage resistance.

In this paper, silt and solidified silt material were added to matrix asphalt. Then, crumb rubber (CR) was employed for composite modification. A parametric study was performed for different contents and sludge solidification. The physical properties and rheological properties were further explored by indicators of penetration, ductility, softening point, and viscosity. Next, we used TFS and LAS to evaluate the rheological properties of the compound-modified asphalt. The water damage resistance and adsorbability were measured by using a Contact Angle and Surface Energy approach. Moreover, the new chemical functional groups or interaction mechanisms between different kinds of concrete, such as CR (crumb rubber), crack, and silt, were determined by virtue of Fourier Transform Infrared Spectroscopy (FTIR) and Fluorescence Microscopy (FM). Using the results gained from the above-mentioned test methods, we can provide detailed guidance regarding the use of Silt/Crumb Rubber Compound-Modified Asphalt (SILT-CRMA) and Composite-modified asphalt with 6% silt content and a solidified silt/cement ratio of 6:4, i.e., 6%HCS (6:4)-CRMA, in practical engineering applications and use our results to inform future studies.

## 2. Materials and Methods

### 2.1. Raw Materials

In this paper, matrix asphalt (70# Gaotong), cement (P.O.42.5), 60-mesh CR, and silt powder were adopted as raw materials. Silt sediment was obtained from the construction site near the S5 contract section of Ningbo Phase I of the double track of Hangzhou-Ningbo Expressway P.O.42.5 (provided by CONCH). The main physical properties of the above-listed substances are provided in Table 1, Table 2, Table 3 and Table 4.

### 2.2. Preparation of SILT-CRMA

Firstly, the dried CR powder (about 10% of the asphalt) was mixed with the asphalt binder at 180 °C, and then stirred in a high-speed shear agitator for 40 min at a shear rate of 5000 r/min. Secondly, the mixture was treated by shear force for 20 min after the addition of various contents of sediment and hardened sediment. Finally, a kind of Silt/Crumb Rubber Compound-Modified Asphalt (SILT-CRMA) was developed, with its expansion and exhaust completed by manual stirring for 30 min at a temperature of 160 °C. Table 3 summarizes the modifiers used in the production of composite-modified tarmac adhesives. At room temperature, the silt and cement were mixed and stirred evenly at a ratio of 6:4. Then, upon adding the appropriate amount of water for curing, the mixture was dried in an oven at 60 °C until constant weight. Eventually, the mixture was crushed into pellets. The microstructure of solidified silt particles obtained after hydration is shown in Figure 1. Table 5 shows the asphalt mixture samples used in this paper. The first four samples are rubber composite-modified asphalt with different sludge contents, while the last two samples are composite-modified asphalt with 6% silt content, in which silt was solidified with cement at a ratio of 6:4 (cement/silt) and unsolidified, respectively.

### 2.3. Testing Methods

#### 2.3.1. Standard Tests

The standard physical properties of various CRMA were tested using penetration at 25 °C, softening point, and ductility at 5 °C, based on ASTM D5, ASTM D36, and ASTM D113, respectively.

#### 2.3.2. Viscosity Test

CRMA was tested for rotational viscosity at 135 °C, 150 °C, 165 °C, and 180 °C using a Brookfield rotational viscometer with the #21 rotor in accordance with ASTM D4402. Different shear rates were changed to ensure that the torque was in the range of 10% to 98%. Viscosity values were measured at different temperatures. Each sample’s viscosity was determined using three replicates. In addition, with the help of the double logarithm method, the relationship between the viscosity–temperature characteristics and thermophysical parameters was obtained, as given in Formula (1).
(1)lg⁡(η)=n−m×lg⁡(T)

In this equation, η denotes the sample’s viscosity at a specific temperature (Pa·s); *T* indicates test temperature (°C); n and m denote the fitting parameters.

#### 2.3.3. Contact Angle Test

Contact angle experiments were performed to estimate the resistance of modified asphalt to water damage. At an ambient temperature of 25 °C, the contact angle of water on modified asphalt was measured using a contact angle meter. Specifically, the asphalt sample was firstly placed on the objective stage of the contact angle tester and the droplets were formed by manual control knob at the interface of the titration needle so that the droplets could drop vertically into the asphalt sample. Finally, we used Equations (2) and (3) to calculate the surface energy of compound modified asphalt [24].
(2)$$\gamma_{s}-\gamma_{sl}=\gamma_{1}\cos\theta$$
(3)$$\gamma_{s}-\gamma_{sl}=2(\sqrt{\gamma_{l}^{d}\gamma_{s}^{d}}+\sqrt{\gamma_{l}^{p}\gamma_{s}^{p}})-\gamma_{l}$$

θ is contact angle (°); $\gamma_{s}$ is asphalt surface energy (mJ/m^2^); $\gamma_{sl}$ is asphalt-liquid surface energy (mJ/m^2^); $\gamma_{l}$ is liquid surface energy (mJ/m^2^); $\gamma_{l}^{d}$, $\gamma_{s}^{d}$ is surface free energy of asphalt and liquid (mJ/m^2^); $\gamma_{l}^{p}$, $\gamma_{s}^{p}$ is polar component of asphalt and liquid (mJ/m^2^).

#### 2.3.4. TFS Test

Several thermoplastic experiments of this new type of cementitious material were carried out with seven frequencies (50, 30, 10, 5, 1, 0.5, and 0.1 Hz) to properly measure its viscoelasticity. Based on the principle of time–temperature equivalence and TFS experimental data, the master curves of the complex modulus (|*G**|) of asphalt binders were drawn [25]. Furthermore, the Christensen–Anderson model (CAM) was numerically simulated using Equations (2) and (3) and the shift coefficient was calculated to acquire the main characteristics of this cementitious material, with 25 °C fitted as the reference temperature. (TR, 25 °C).
(4)$$|G^{*}|=|G_{g}^{*}|{\left[1+{\left(\frac{\omega_{c}}{{\omega \alpha }_{T}}\right)}^{\left(v\right)}\right]}^{-w/v}$$
(5)$$\log\left(\alpha_{T}\right)=\alpha_{1}{\left(T-T_{R}\right)}^{2}+\alpha_{2}\left(T-T_{R}\right)+\alpha_{3}$$

As shown above, glassy modulus is taken as 1 or 1.2 GPa; mean crossover angular frequency and physical angular frequency refer to dimensionless shape parameters of the master curves and regression coefficients, respectively.

#### 2.3.5. LAS Test

According to the American AASHTOT101 code, the stress amplitude of the LAS test increased from 0.1% to 30% at 25 °C at a stable frequency of 10 Hz. Based on the simplified viscoelastic-continuous failure (S-VECD) theory, the experimental results of LAS were statistically analyzed and discussed using DCC (Density Conference) method. DCC can reveal the internal relationship between the overall properties (virtual rigidity, C) and the damage degree (S) of materials under full working conditions to effectively predict fatigue performance. Its performance is independent of the time and environment of the load. Equations (4) and (5) present the dynamic elastic modulus rate (DMR) and intrinsic damage evolution variable (S (t)), respectively, and Equations (6)–(9) offer the simulated virtual strain value (*N_f_*).
(6)$$DMR={\left|G^{*}\right|}_{Figerprint}/{\left|G^{*}\right|}_{LVE}$$
(7)$$S\left(t\right)=\sum_{i=1}^{n}[\frac{DMR}{2}{{\left(\gamma_{P,i}^{R}\right)}^{2}(C_{i-1}^{*}-C_{i}^{*})]}^{\frac{\alpha }{\alpha +1}}\cdot {\left(t_{Ri}-t_{Ri-1}\right)}^{\frac{1}{1+\alpha }}$$
(8)$$C^{*}\left(S\right)=1-C_{1}{\left(S\right)}^{C_{2}}$$
(9)$$N_{f}={\left(\frac{A}{a}\gamma_{P}^{2+\frac{2\alpha C_{2}}{v}}\right)}^{\frac{1}{b+1-C_{2}/v}}$$
(10)$$A=\frac{1}{2}C_{1}{\left({|G}^{*}|\right)}^{2}B^{-\frac{C_{2}}{1-\alpha C_{2}+\alpha }}\frac{1}{1+C_{2}/(1-\alpha C_{2}+\alpha )}$$
(11)$$B=\frac{f_{R}2^{\alpha }}{(1-\alpha C_{2}+\alpha )(C_{1}C_{2}{)}^{\alpha }(|G^{*}|{)}^{2\alpha }}$$

In these equations, $t_{R}$ represents the reduced time; i refers to the cycle number; $\Upsilon_{\rho }^{R}$ indicates nonlinear pseudo strain; α denotes a material constant in a non-damaged state; *n* stands for the maximum number of loading times; |*G**|Fingerprint and |*G**|LVE refer to the measured initial |*G**| and linear viscoelastic |*G**|, respectively; *C*_1_ and *C*_2_ are optimized fitting parameters of solidified silt.

#### 2.3.6. FTIR Test

SILT-CRMA functional groups were studied using FTIR in the wavelength range of 450–4000 cm^−1^, with spectra captured at a resolution of 5 cm^−1^.

#### 2.3.7. FM Test

FM (LF-50) was utilized to observe the polymer phase structure of the composite-modified asphalt binder. For the FM test, a small previously prepared bitumen sample was placed on a heated glass sheet and evenly spread on a glass slide at 400× *g* magnification.

## 3. Results and Discussion

### 3.1. Conventional Physical Properties

In the ordinary asphalt property test, penetration, ductility, and softening point are the most common parameters. Figure 2 shows that, in different concentrations of clay, the permeability coefficient of clay decreases, but its softening point shows an upward trend. Specifically, the temperature stability of ceramics is better when the mass fraction of powder is 8%. Mechanical mixing of cementitious materials with silty clay can lead to a reduction in the penetration value and an increase in the softening point, whereas the hardening treatment with cement slurry can increase its penetration value and decrease its softening point, thus making it thicker and maintaining good stability at higher temperatures. Under the condition of different doping amounts, the ductility value decreases with the increase in doping amount. However, after hardening treatment with cement slurry, the AFT value increases slightly. On the interface between AFT and asphalt, the ductility of modified asphalt can be improved by adding a weak contact point for achieving stronger anti-plasticity.

The performance of different modification methods is demonstrated by the order of the following sequence: 6%HCS (6:4)-CRMA > 6%UHCS (6:4)-CRMA > 8%SILT-CRMA > 6%SILT-CRMA > 4%SILT-CRMA > 2%SILT-CRMA.

### 3.2. Viscosity

Figure 3 depicts the double logarithmic curves of SILT-CRMA viscosity at various temperatures. The incorporation of CRMA increases the viscosity, and the viscosity of CRMA is enhanced to a more significant extent after the solidification of silt. When silt and cement are blended physically at a ratio of 6:4, the viscosity of asphalt increases [26]. Moreover, the viscosity of modified asphalt is significantly improved when the silt is solidified by cement. As expected, as the silt content changes, CRMA can withstand a high temperature stress—even higher when cement-solidified slit is added. Apart from that, the experimental results also demonstrate that the rotational viscosity of SILT-CRMA decreases with the increase in temperature, which is consistent with the non-Newtonian flow under experimental conditions. However, the rate of viscosity reduction is extremely small under the higher powder mass silt and higher test temperature. When the water temperature is maintained at a low temperature, the effect of sediment concentration is more significant. The viscosity of 6% HCS (6:4)-CRMA increases by 48.2% and 24.5% at 135 °C and 180 °C, respectively, compared to 2% SILT-CRMA. Furthermore, the viscosity of 6% HCS (6:4)-CRMA is greater than that of 6% UHCS (6:4)-CRMA, which, to a certain extent, is due to changes in the interactions between the components of the solidified silt.

For asphalt binders, the mixing pressure should be 0.17 Pa·s ± 0.02 Pa·s, and the compaction pressure should be 0.28 Pa·s ± 0.03 Pa·s. The results for the mixing and compaction temperatures of SILT-CRMA are described in Figure 4. It can be seen that the higher the content, the lower the mixing and compaction temperatures. The mixing and compaction temperatures of 6% HCS (6:4)-CRMA were the highest. Although higher viscosity means that higher mixing and compaction temperatures are required for construction, the measured viscosity values conform to the Superpave standard (≤3 Pa·s at 135 °C) [27]. In addition, the mixing and compaction temperatures of 6% UHCS (6:4)-CRMA are higher than those of 6% SILT-CRMA, indicating that the addition of cement increases the viscosity of the modified asphalt. Our research results on the different mixing and compaction temperatures of SILT-CRMA and HCS-CRMA can be used as guides in the field of practical construction.

### 3.3. Temperature Frequency Scanning

In Figure 5, the master curve of the composite elastic coefficients of different kinds of SILT-CRMA are drawn. |*G**| represents the total resistance subjected to the repeated shearing of the asphalt binder. As shown in ref. [28], |*G**| also gradually increases when reduced frequency increases. This indicates that low temperatures and high frequencies can bring about a significant increase in the resistance of SILT-CRMA, in the case of repeated shearing. From Figure 5, it can be seen that |*G**| gradually increases with the rise in silt content. The silt content reaches 6% when |*G**| is the largest. However, when the silt content is 8%, the resistance of the modified asphalt declines to a certain extent. When the cement and silt are only physically blended, more air voids will appear in cement and silt, which can result in a significant decrease in the |*G**| value of the modified asphalt. |*G**| can reach the best value when the silt is solidified with cement. The main reason behind this is due to the reticular structure of cement mortar filled by dense and hardened silt [29]. Given the physical blending of cement and silt, large porosities emerged in the cement and silt, which can lead to a reduction in the |*G**| value of the modified asphalt.

### 3.4. Linear Amplitude Scanning

At 25 °C, the LAS performance of the modified asphalt cementitious material with different amounts of cement hardening mixture was studied [30]. As displayed in Figure 6, the shear stress of asphalt appears to have a peak value, indicating the occurrence of an inflection point. Additionally, the results that indicate a decrease can be attributed to the failure of asphalt due to the increase in shear stress. Damage can be observed in 2% SILT-CRMA when the strain reaches 14%. When an appropriate amount of silt is replaced with cement, the strain of the modified asphalt changes to 14.5%, the width of the peak stress–strain zone increases, and the sensitivity to strain is reduced to a certain extent. Furthermore, the curves of 6% HCS (6:4)-CRMA are different from those of the other asphalt samples. When the silt is solidified by cement, its initial failure occurs at a higher strain than the other asphalt samples. The width of the peak zone of the stress–strain curve is larger compared to other modified bitumen, which indicates that 6% HCS (6:4)-CRMA is less sensitive to strain than other modified bitumen and capable of producing excellent anti-fatigue performance.

Based on the S-VECD model, the DCC model is used to characterize the internal failure of asphalt concrete [31]. In Figure 7, the horizontal coordinate D refers to the cumulative failure coefficient, and the longitudinal coordinate C represents the overall performance coefficient of asphalt samples. It can be observed from the chart that the overall performance of cement paste decreases sharply, showing a continuous increase in terms of damage. The C*(S) curve of 2% SILT-CRMA declines faster, indicating that it has a worse fatigue resistance. With the increase in silt content, the fatigue resistance of modified asphalt was improved. Comparing 6% HCS (6:4)-CRMA and 6% UHCS (6:4)-CRMA, it can be found that the fatigue resistance of modified asphalt can be significantly improved after the appropriate amount of silt is solidified. However, the main factors affecting DCC are destruction factors (C_1_ and C_2_), which generate great fluctuations on account of the difference in sediment concentration and the complexity of material composition. As a consequence, in some developing countries, this direction of development is basically the same, which suggests that there are certain obstacles inhibiting the ability of researchers to evaluate the durability of SILT-CRMA by DCCS alone [32,33].

Under various stress levels (2.5%, 5%, 10%), Figure 8 is fitted by the logarithm method. The results show that the slope of the formula is sensitive to stress. It can be concluded that the greater the content of cement stone, the greater the shear strength of concrete. Nevertheless, the momentum of growth weakens gradually. It can be seen that excessive sludge is disadvantageous to enhance the fatigue resistance of asphalt, which is mainly due to the aggregation effect of modifiers. Its fatigue resistance can be improved by hardening with cement slurry.

### 3.5. Contact Angle Test

The SILT-CRMA contact angle of different silt contents is shown in Figure 9. The contact angle indicates the compatibility between these two materials. The larger the contact angle, the worse the performance in wet tability. With the increase in silt content, the contact angle between asphalt and water progressively decreases, suggesting that the contact between asphalt and water becomes closer, contributing to the asphalt’s reduced capability to resist water damage. During the physical blending of silt and cement, the contact angle of 6% UHCS (6:4)-CRMA gradually declines. However, the contact angle between modified asphalt and water increases significantly after the silt is solidified by cement. The contact angle between 6% HCS (6:4)-CRMA and water reaches 92°, which makes the modified asphalt hydrophobic. In summary, 6% HCS (6:4)-CRMA performs well in terms of water damage resistance.

In Figure 10, it can be found that the surface energy of SILT-CRMA gradually decreases with the increase in SILT content; thus, the energy required to form a new interface between compound-modified asphalt and aggregate is reduced. When the silt and cement are only physically blended, the surface energy of the modified asphalt will be slightly increased. However, when the silt is solidified by water and cement, the surface energy of the modified asphalt will be significantly reduced. The smaller the surface energy of modified asphalt, the harder it was for the asphalt to absorb water molecules. Therefore, compound-modified asphalt has better water damage resistance. In summary, 6% HCS (6:4)-CRMA performs well in terms of water damage resistance.

### 3.6. Fourier Transform Infrared Reflection

For the present study, FTIR was adopted to study the functional groups of SILT-CRMA, and the spectrum of different SILT-CRMAs is illustrated in Figure 11. It should be pointed out that, in these basic functions, the presence of tarmac cementing materials is obvious. Through observing the spectral curves below, it can be found that the peak absorptivity of medium and high density around 1400 cm^−1^ is caused by-CH_2_-, while the peak absorptivity of medium and high density around 1350 cm^−1^ is caused by-CH_3_. Additionally, there are large absorption waves at the peaks of 2920 cm^−1^ and 2850 cm^−1^, indicating that the asymmetric stretching motion of methylene carbon-hydrogen bonds occurs. However, a medium-strong C-H shear vibration absorption peak appears at 1485 cm^−1^. Around 860 cm^−1^, a weak absorption wave indicates the buckling oscillation of the benzene ring on the C-H bond. The peaks at 725 cm^−1^ and 1000 cm^−1^ represent the extensional motion of C=C and C-H in the 4-butadiene skeleton, and the peak at 810 cm^−1^ reveals the out-of-plane buckling motion of the group C-H in the isoprene skeleton. These three functions generally refer to CR. The functional groups in clay and cement cannot be reflected in conventional IR spectra. Therefore, there is no new chemical bonding between cement, CR, and asphalt cementitious materials, according to the obtained spectral curves. In other words, silt, cement-stabilized macadam, and pavement asphalt are a kind of solid mixture.

### 3.7. Fluorescence Microscope

As shown in Figure 12, FM could be used to intuitively observe the dispersion of SILT-CRMA, which could also be detected by using fluorescence microscopy. The black and bright parts in the middle represent the matrix asphalt and rubber particles, respectively. Nevertheless, silt can generally be analyzed indirectly via matrix asphalt since it has no fluorescence effect. Because of the frequency modulation of luminosity in separate batches, the asphalt phase has different background colors. According to Figure 12a–d, the rubber particles are more evenly distributed. With the increase in silt content, the black asphalt matrix agglomeration (area) is significantly expanded, which may be because the porosity of silt soil particles is conducive to adsorbing matrix asphalt, leading to the deterioration of compatibility in the asphalt phase. The physical mixing of silt and cement can help generate more pores between cement and silt to adsorb asphalt, resulting in a significant increase in the area of matrix asphalt [34,35]. As displayed in Figure 12e, however, the stability of the solidified silt material, rubber, and asphalt is the highest after the silt is fully solidified by cement. The reason for this may be that the silt solidified by cement will form cementing materials between soil particles and increase the degree of cementation between particles, which ultimately causes significant contraction in the area of matrix asphalt.

## 4. Conclusions

In this paper, the influences of different silt contents on the physical chemical composition and properties of composite-modified asphalt binder are investigated. The conclusions drawn based on the experimental data can be briefly summarized as follows:

(1) The permeability and toughness of the modified asphalt binder decrease with the increase in silt content, while its softening point increases. 6%SILT−CRMA has superior conventional performance properties, whereas, after the silt is solidified by cement, both penetration and ductility are clearly improved, and the softening point shifts downwards slowly. The rotational viscosity of SILT−CRMA decreases with the increase in experimental temperature, and the mixing and compaction temperatures of 6% UHCS (6:4)−CRMA are higher than those of 6% SILT−CRMA. Furthermore, the rise in viscosity results in higher mixing and compaction temperatures, satisfying the requirements of Superpave specifications.

(2) With the increase in silt content, the SILT−CRMA contact angle gradually decreases, surface energy slowly declines, and the resistance of the modified asphalt to moisture damage is reduced. However, resistance is clearly enhanced after the addition of solidified silt material.

(3) 6% HCS (6:4)−CRMA outperforms other types of modified asphalt in terms of high-temperature stability and fatigue resistance. However, adding silt or silt and cement mixture alone is not conducive to improving the performance of compound-modified asphalt. The dense structure formed by the cement-based solidification of silt can effectively improve the asphalt matrix’s rheological properties and water damage resistance.

(4) From the thermography of FTIR and FM, it can be concluded that 6% HCS (6:4)−CRMA is similar to SILT−CTMA in terms of composition and structure, which signifies that the addition of silt and solidified silt is only physical blending. The mixture of solidified silt, rubber, and asphalt did not produced new chemical functional groups to break the chemical structure of the asphalt.

(5) After the asphalt matrix is modified by solidified slit and CR, the rheological properties and fatigue resistance of HCS−CRMA are generally superior to that of SILT−CRMA.

## Figures and Tables

**Figure 1 polymers-15-02474-f001:**
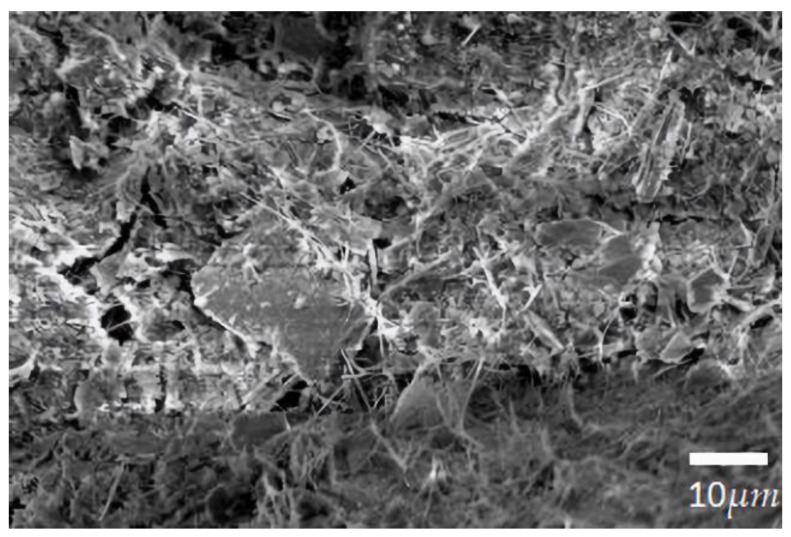
The morphology of solidified silt.

**Figure 2 polymers-15-02474-f002:**
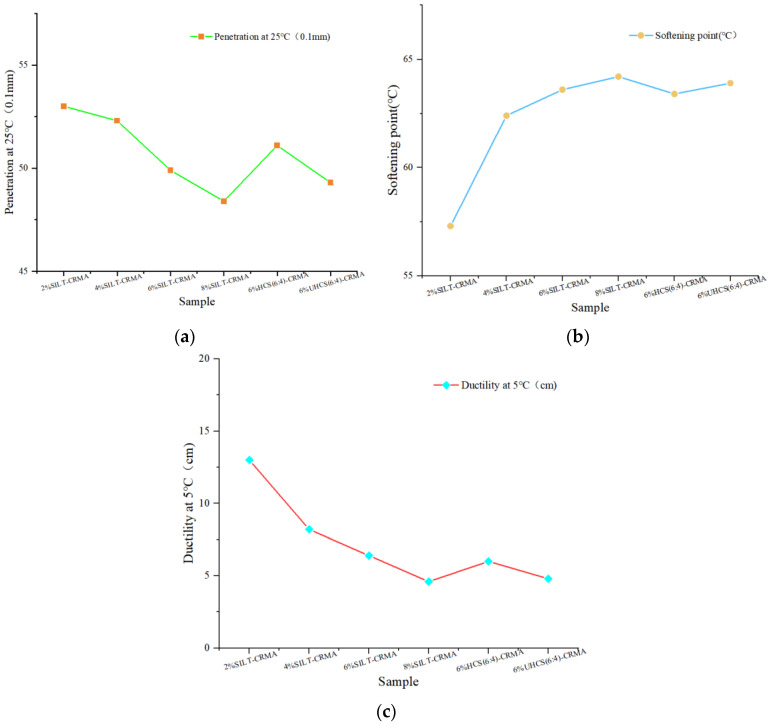
Effect of SILT and CEMRNT on conventional physical properties. (**a**) Penetration (25 °C, 0.1 mm); (**b**) Softening point (°C); (**c**) Ductility (15 °C, cm).

**Figure 3 polymers-15-02474-f003:**
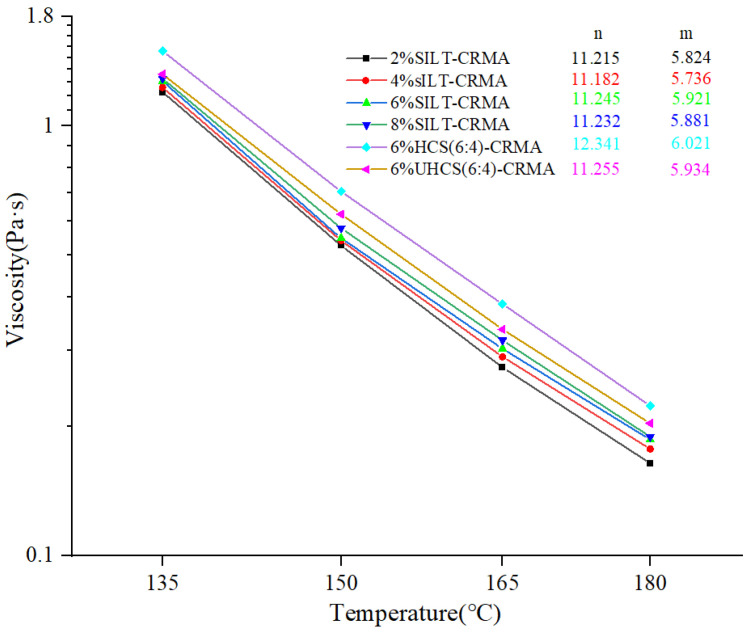
Viscosities of SILT-CRMA at various test temperatures.

**Figure 4 polymers-15-02474-f004:**
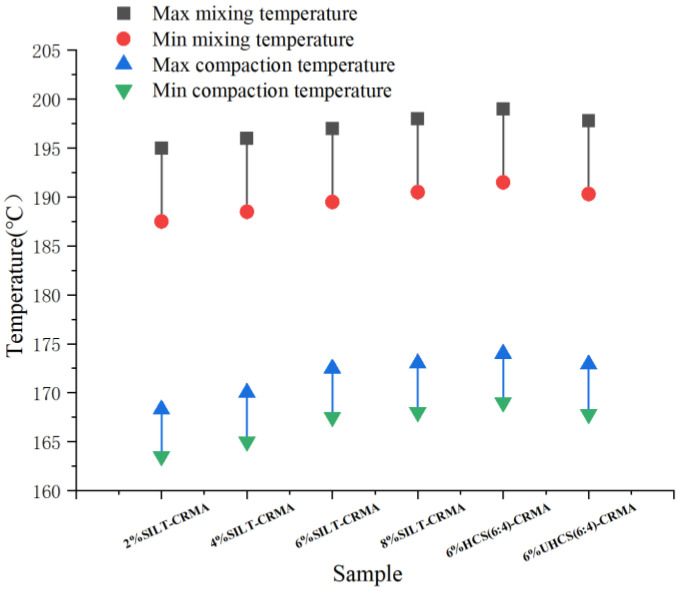
Mixing and compaction temperatures of SILT-CRMA.

**Figure 5 polymers-15-02474-f005:**
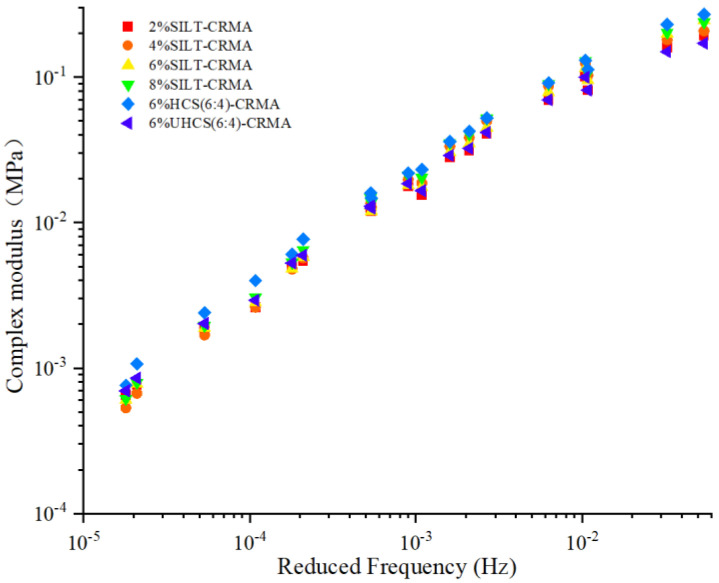
Complex modulus master curve of different SILT−CRMA.

**Figure 6 polymers-15-02474-f006:**
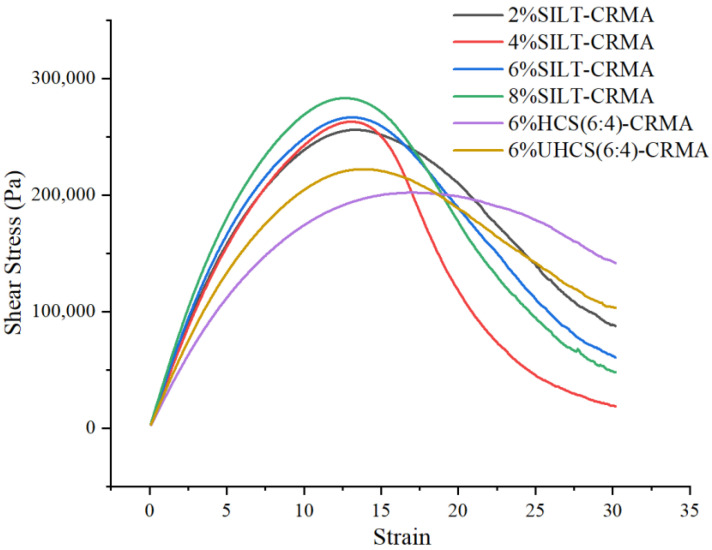
Stress–strain curves of SILT-CRMA.

**Figure 7 polymers-15-02474-f007:**
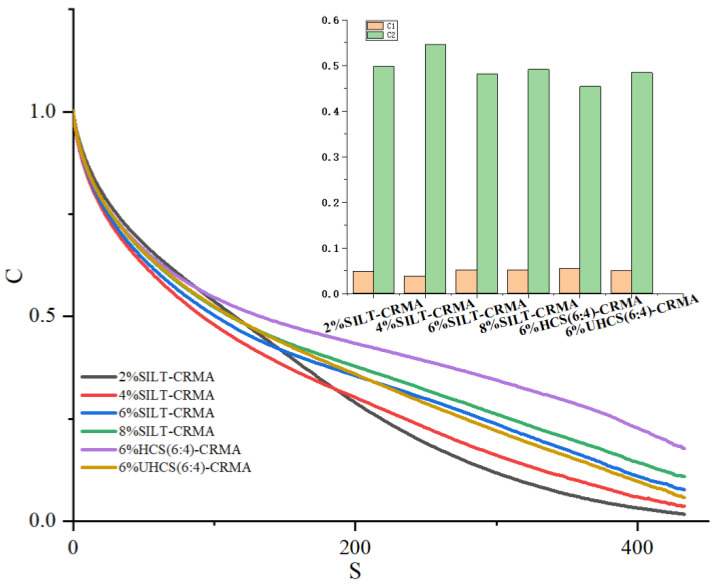
Damage characteristic curves of SILT−CRMA.

**Figure 8 polymers-15-02474-f008:**
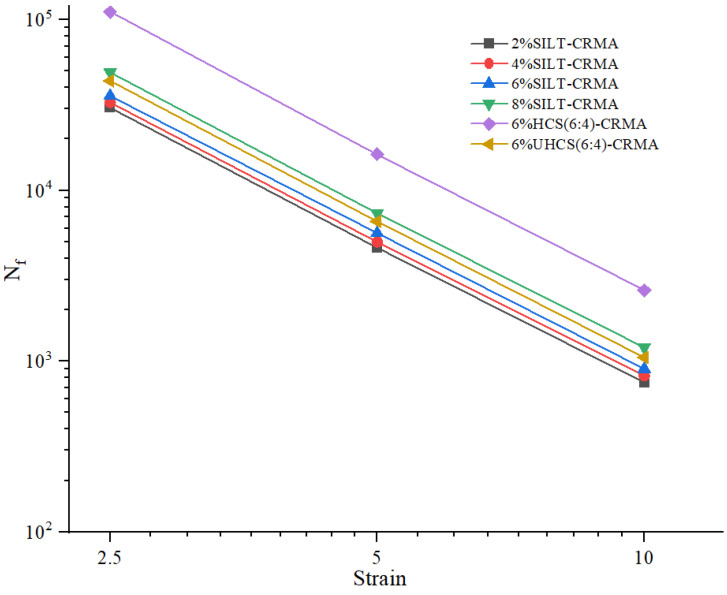
Fatigue resistance of SILT−CRMA.

**Figure 9 polymers-15-02474-f009:**
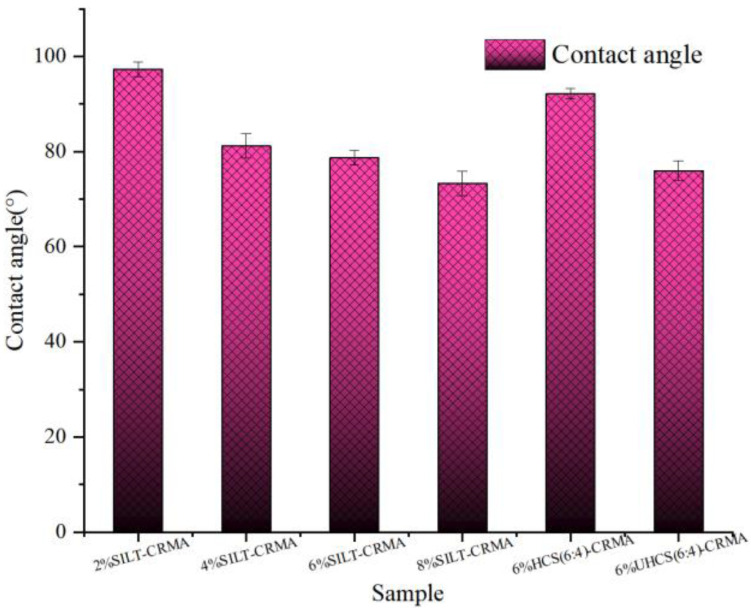
Contact angle of SILT−CRMA.

**Figure 10 polymers-15-02474-f010:**
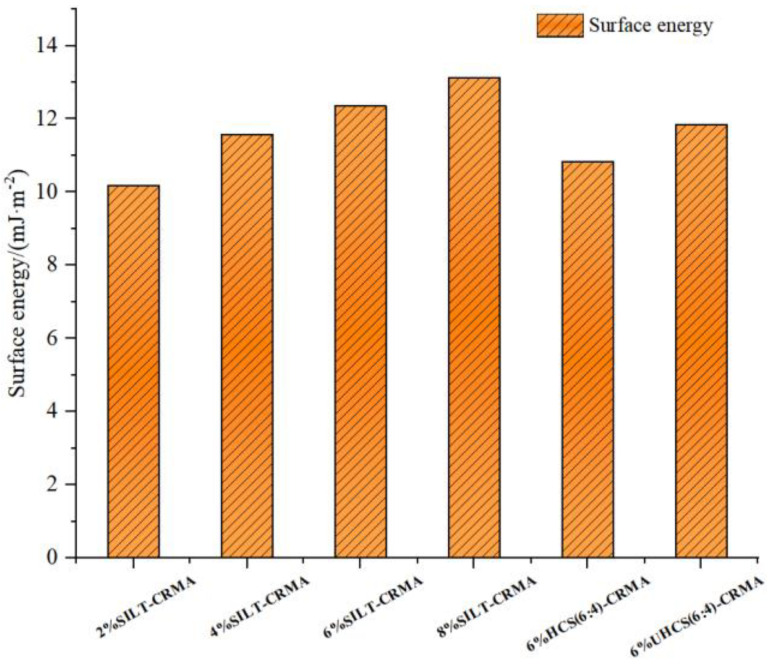
Surface energy of SILT−CRMA.

**Figure 11 polymers-15-02474-f011:**
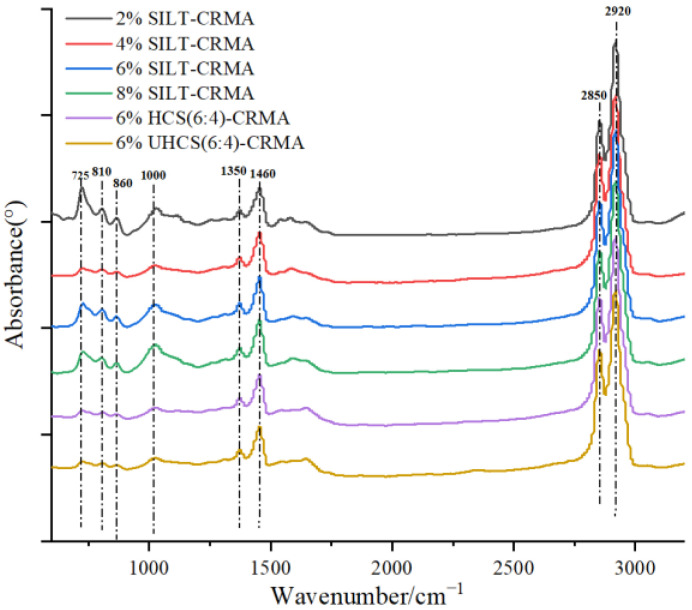
The spectrum of SILT-CRMA.

**Figure 12 polymers-15-02474-f012:**
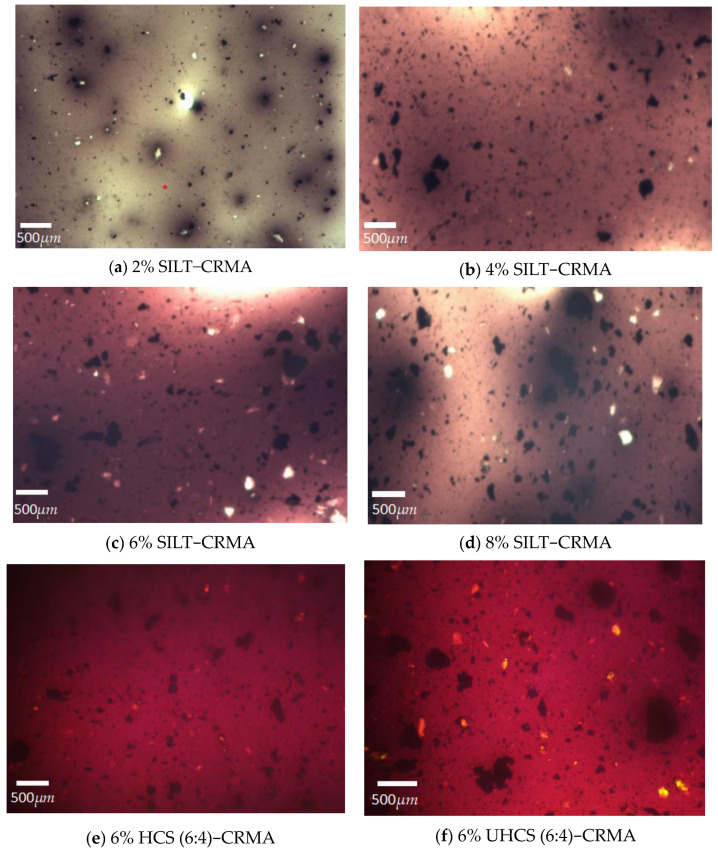
The microstructure of SILT-CRMA.

**Table 1 polymers-15-02474-t001:** Physical properties of matrix asphalt.

Brand	Softening Point (°C)	Penetration (25 °C, 0.1 mm)	Ductility (15 °C, cm)
Gaotong	47.1	65.6	>100

**Table 2 polymers-15-02474-t002:** Physical properties of Silt.

Moisture Content (%)	Cavity Ratio	High Liquid (%)	Plastic Limit (%)	Plastic Index	Compression Coefficient	Soil Natural Density (g·cm^−3^)	Cohesive Force (kPa)	Organic Content (%)
45.1	1.17	39.1	14.3	24.8	0.76	1.76	11.8	1.78

**Table 3 polymers-15-02474-t003:** Physical properties of cement.

Technical Index		Code Value	Measurements
Fineness		≥300 m^2^/kg	310 m^2^/kg
Soundness		Conformity	Conformity
Setting Time	Initial set Final set	≥45 min	100 min
		≤600 min	190 min
Compressive strength	3 d 28 d	≥17.0 Mpa	20 Mpa
		≥42.5 Mpa	47.5 Mpa

**Table 4 polymers-15-02474-t004:** Physical properties of Crumb Rubber.

Property	Relative Density	Fiber Content (%)	Water Content (%)	Metal Content (%)
Test value	1.25	0.3	0.35	0.04
Screening test				
Trough rate (%)	95.4	2.7	65.8	0.55
Sieve size (mm)	0.7	0.2	0.25	0.08

**Table 5 polymers-15-02474-t005:** Contents of compound modifiers and abbreviation of asphalt samples.

Mix Cement and Silt	Silt Content (%)	Cement Content (%)	CR Content (%)	Abbreviation
0	2	0	10	2%SILT-CRMA
0	4	0	10	4%SILT-CRMA
0	6	0	10	6%SILT-CRMA
0	8	0	10	8%SILT-CRMA
6 (Hydration)	Cement:Silt (6:4)		10	6%HCS (6:4)-CRMA
	Cement:Silt (6:4)		10	6%UHCS (6:4)-CRMA

## Data Availability

In order to reproduce this result, the necessary raw data that are part of this work cannot be shared currently.

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
