# Peer review of "Investigation into the Rheological Properties and Microstructure of Silt/Crumb Rubber Compound-Modified Asphalt"

_polymers, 2023, doi:10.3390/polym15112474_

Round 1

Reviewer 1 Report

The article is interesting for scientific community, but there are aspects to improve before the article publication:

Need to explian and introduce more deatils about how has been performed Damage characteristic curves of SILT-CRMA.

Titles of subsections can not be acronyms, needs to include more detailed description (subsections: LAS, TFS, FM...

You need to justify the tests performed in the research according with the goals of the article, please introduce and include this part

In conclusions you need to justify the use of the tested materials for specific applications, and the possibilities of use the tested materials in applications and explain the key characteristics of tested materials.

Author Response

Response to Reviewer 1 Comments

Dear reviewer,

First and foremost, many thanks for your time and comments concerning our manuscript. Those comments are all valuable and very helpful for helping us revise and improve the paper. We have studied the comments carefully and made revisions to the text which are correspondingly highlighted in red in the revised manuscript. We hope they can meet your standard. The detailed responses to the specific points raised are given in the following paragraphs.

Point 1: Need to explian and introduce more deatils about how has been performed Damage characteristic curves of SILT-CRMA. 

 Response 1: Dear reviewer, thank you for your suggestion. We added a more detailed explain and introduce. 

“The C*(S) curve of 2% SILT-CRMA moves down faster, indicating its worse fatigue resistance. With the increase of silt content, the fatigue resistance of modified asphalt was improved. Comparing 6% HCS (6:4)-CRMA and 6% UHCS (6:4)-CRMA,we can found that the fatigue resistance of modified asphalt can be significantly improved after the appropriate amount of silt is solidified.”

Point 2:Titles of subsections can not be acronyms, needs to include more detailed description (subsections: LAS, TFS, FM...

Response 2: Dear reviewer, thank you for your suggestion. We have amended similar problems in this paper.

“ 3.3.Temperature Frequency Scanning; 3.4.linear Amplitude Scanning; 3.5. Contact Angle test; 3.6. Fourier Transform Infrared Reflection; 3.7. Fluorescence Microscope”

Point 3:You need to justify the tests performed in the research according with the goals of the article, please introduce and include this part

Response 3: Dear reviewer, sorry for not explaining clearly. We have added some explain to introduction part.

“Next, we have used TFS and LAS to evaluate the rheological properties of the compound modified asphalt. The water damage resistance and adsorbability were texted by Contact Angle and Surface Energy. By the above test method, to provided detail guidance for the Silt/ Crumb Rubber Compound Modified Asphalt in the practical engineering application and to be further studied.”

Point 4:In conclusions you need to justify the use of the tested materials for specific applications, and the possibilities of use the tested materials in applications and explain the key characteristics of tested materials.

Response 4: Dear reviewer, thanks for your advice. We have added some explain to abstract part.

“Therefore, 6%HCS(6:4)-CRMA have better performance. Compared to ordinary rubber modified asphalt, the composite modified asphalt binder has better physical properties and more suitable construction temperature. The composite modified asphalt uses discarded rubber and silt as raw materials, which can effectively protect the environment. Meanwhile, the modified asphalt has excellent rheological property and fatigue resistance.”

Thank you very much again for your comments and suggestions. We tried our best to improve the manuscript and made some changes in the revised manuscript. We appreciate for your warm work earnestly and hope that the correction will meet your approval.

Best wishes,

Lu Huang

Reviewer 2 Report

 The paper try to find a recycling solution for the silt (construction waste) with mixing (<8%) it with CR (10%) and asphalt. In order to evaluate the feasibility of the proposal, it is carried out a study on the rheological properties of the asphalt SILT-CRMA. The paper is clear and well structured. In my opinion the introduction could be improved by listing all the acronims used. The results and discussion  could be also be improved by given a table in whitch was summarired the main propierties of SILT-CRMA compared with an CRMA in order to clearly present the advatades/disadvantages of the 6 samples estudied.

Author Response

Response to Reviewer 2 Comments

Point 1: In my opinion the introduction could be improved by listing all the acronims used.

Response 1: Dear reviewer, thank you for your suggestion. We have changed all the acronym in the introduction part.

“In this paper, the silt and silt solidified material were added to matrix asphalt, and then (crumb rubber)CR was employed for composite modification. A parametric study was performed for different contents and sludge solidification. The physical properties and rheological properties were further explored by indicators of penetration, ductility, softening point, viscosity and temperature as well as methods including Frequency Scanning (TFS), Linear Amplitude Scanning (LAS) and Contact angle test.The water stability of the compound modified asphalt can be evaluated by the Contact angle and Surface energy measured in the contact angle experiment. Moreover, the new chemical functional group or interaction mechanism between different kinds of concrete such as CR, crack and silt was discussed profoundly by virtue of Fourier Transform Infrared Reflection(FTIR) and Fluorescence Microscope(FM). “

Point 2: The results and discussion could be also be improved by given a table in whitch was summarired the main propierties of SILT-CRMA compared with an CRMA in order to clearly present the advatades/disadvantages of the 6 samples estudied.

Response 2: Dear reviewer, sorry for not explaining clearly. Thus, in order to better explain the advantages and disadvantages of SILT-CRMA and HCS-CRMA ,we provide supplementary explanations in the conclusions part.

“The rotational viscosity of SILT-CRMA system decreases with the increase of experimental temperature and the mixing and compaction temperatures of 6% UHCS (6:4)-CRMA are higher than those of 6% SILT-CRMA.

With the increase of silt content, the SILT-CRMA contact angle gradually decreases and surface energy slowly decline, and the resistance of the modified asphalt to moisture damage is reduced.

The dense structure formed by cement solidification of silt, it can effectively improve the rheological property and water damage resistance of asphalt matrix.

The mixture of silt solidification, rubber and asphalt didn't produced new chemical functional groups to break the chemical structure of the asphalt.”

Thank you very much again for your comments and suggestions. We tried our best to improve the manuscript and made some changes in the revised manuscript. We appreciate for your warm work earnestly and hope that the correction will meet your approval.

Best wishes,

Lu Huang

Reviewer 3 Report

After reading the submitted manuscript I am not sure whether this is the final draft or not. Referencing to Figures instead of Tables (l. 104), bad numeration of table references (l. 115), yellow blocks in the manuscript, incomplete inscription of the parameters (l. 157).

The manuscript is not well arranged and orientation is really difficult caused among other things by using numerous abbreviations without any explanations. At many places the referencing is completely missing thus providing the readers no idea about the discussed topics.

The headline ´2. Materials and Devices´ is also missing, the used devices are not properly denoted (l. 132 as one out of many examples), the titles of the subsections are insufficient.

Many statements are rather difficult and there is no idea what these statements should represent as e.g. temperature in Pa.s (l. 218) of the term double logarithmic curves (l. 202 if the semi-log coordinates were used).

In the section ´Conclusions´ there is a mere description of the individual figures with no general recommendation or analysis.

Based on the above points I cannot recommend this manuscript for publication in the journal Polymers as quality does not fulfil a standard of this Journal.

Author Response

Response to Reviewer 3 Comments

Dear reviewer,

First and foremost, many thanks for your time and comments concerning our manuscript. Those comments are all valuable and very helpful for helping us revise and improve the paper. We have studied the comments carefully and made revisions to the text which are correspondingly highlighted in red in the revised manuscript. We hope they can meet your standard. The detailed responses to the specific points raised are given in the following paragraphs.

Point 1: The headline 2. Materials and Devices´ is also missing, the used devices are not properly denoted (l. 132 as one out of many examples), the titles of the subsections are insufficient.Many statements are rather difficult and there is no idea what these statements should represent as e.g. temperature in Pa.s (l. 218) of the term double logarithmic curves (l. 202 if the semi-log coordinates were used).

Response 1: Dear reviewer, sorry for the clerical error on the papers.We have revised similar format problem and reorganize the language to describe.

“2. Materials and methods; 3.3.Temperature Frequency Scanning; 3.4.linear Amplitude Scanning; 3.5. Contact Angle test; 3.6. Fourier Transform Infrared Reflection; 3.7. Fluorescence Microscope;

Point 2: In the section ´Conclusions´ there is a mere description of the individual figures with no general recommendation or analysis.

Response 2: Dear reviewer, sorry for not explaining clearly. We have added more clearly explain in Conclusions part.

“The rotational viscosity of SILT-CRMA system decreases with the increase of experimental temperature and the mixing and compaction temperatures of 6% UHCS (6:4)-CRMA are higher than those of 6% SILT-CRMA.

With the increase of silt content, the SILT-CRMA contact angle gradually decreases and surface energy slowly decline, and the resistance of the modified asphalt to moisture damage is reduced.

The dense structure formed by cement solidification of silt, it can effectively improve the rheological property and water damage resistance of asphalt matrix.

The mixture of silt solidification, rubber and asphalt didn't produced new chemical functional groups to break the chemical structure of the asphalt.”

Thank you very much again for your comments and suggestions. We tried our best to improve the manuscript and made some changes in the revised manuscript. We appreciate for your warm work earnestly and hope that the correction will meet your approval.

Best wishes,

Lu Huang

Reviewer 4 Report

Polymers

Investigation into the rheological properties of silt/crumb rubber compound modified asphalt.

Comments:

This study investigates the rheological, chemical, and microstructure of silt/crumb rubber composite modified asphalt. This topic is interesting and valuable to engineering application with both economic and environmental benefits. Some comments are given to further improve the manuscript.

(1) The chemical and microstructural properties are also included in this paper, why only the rheological properties term is mentioned in the title?

(2) It would better to mention the optimum dosages of silt and crumb rubber modifiers based on the experimental results.

(3) How do the authors think about the moisture influence on the compound asphalt? The high moisture absorption capacity of silt may cause the moisture damage of asphalt moisture. Moreover, the volume expansion of silt under the moisture condition would also show a significant influence.

(4) The introduction section has been improved with more state-of-the-art references. For example: Investigating the role of swelling-degradation degree of crumb rubber on CR/SBS modified porous asphalt binder and mixture. Construction and Building Materials. Extruded tire crumb-rubber recycled polyethylene melt blend as asphalt composite additive for enhancing the performance of binder. Journal of Materials in Civil Engineering.

(5) The information of some references (e.g. [16] [17]) is not correct, pls check and revise it.

(6) The research objectives should be supplemented at the end of Introduction part.

(7) The abbreviation of “TFS test” should be displayed with the full name. Same issue occurs in the LAS test.

(8) The axis scope has to be reorganized to clearly see the difference, for instance Figure 2.

(9) I cannot see the analysis on the viscosity-temperature correlation with Eq.1 in Figure 3.

(10) Based on the Figure 5, no significant difference of G* of various modified samples is detected.

(11) More information of contact angle, such as surface energy, can be provided.

(12) How about the aging and low-temperature properties of compound modified asphalt?

Round 2

Reviewer 3 Report

I am afraid that the changes carried out in the manuscript are insufficient. The reasons are as follows

·      Title: not corresponding to the contents of the manuscript, not only rheological aspects are considered.

·      Introduction: misses compactness. Some abbreviations are not necessary to introduce (FM), some are not explained (HCS, CRMA). as a whole, the Introduction is not well organised.

·      I have to repeat the paragraph from my preceding review: "The manuscript is not well arranged and orientation is really difficult caused among other things by using numerous abbreviations without any explanations. At many places the referencing is completely missing thus providing the readers no idea about the discussed topics." Please, compare with ll.35, 49, 55, 62, 63, 64, 76, etc.

·      Why the reference [17] was eliminated – not to renumber other references?

·      References in the Introduction are cumulated with no arrangement, the results in these references should be more interlaced, otherwise it does not provide a comprehensive outline.

·      The last paragraph in the Introduction does not provide an aim of this manuscript, substantiation of the used methods and the goals which should be achieved. A mere description of the used methods is insufficient.

·      l.121: Fig. or Table?

·      l. 128: What does it mean: at a rate of 180 degrees?

·      l. 132: Table 3 or 5? What is a difference between last two lines in Table 5?

·      Figure 1 does not indicate any scale, hence, it provides no information.

·      l. 145 – referencing?

·      ll. 151, 152 – What is meant by the term "the double-logarithmic method"?

·      l. 150: What does it mean "... the torque was in the range of 10% to 98%"?

·      No geometrical parameters of rheological tests presented.

·      I should repeat my sentence from the preceding review: "... the used devices are not properly denoted, the titles of the subsections are insufficient".

·      l. 173: no reference introduced

·      l. 174: how the equations 2-3were used?

·      ll. 177-179: notation is missing

·      l. 184: Explanation of the abbreviation is shortened

·      l. 203: Prior to the section ´Results and Discussion´ there is so many imperfections that a reader has almost no idea what and how is measured.

·      Just in brief:

-     Fig. 2: the connecting segments have no sense;

-     Fig. 3: the curves do not correspond to rel. (1) because this relation predicts a smooth continuous curve, not a series of segments;

-     Fig. 3: the values of n and m are presented up to five valid ciphers, absolutely in contradiction with the experimental accuracy;

-     Fig. 5: the term ´master curve´ used in the figure caption; however, no master curve is presented both in Fig. 5 and in the text;

-     Fig. 7: illegible description of the inserted picture, insufficient description of both axes;

-     Fig. 8: what does it mean: double logarithmic method?;

-     Fig. 11: no scale presented.

·      Conclusions: really difficult to be oriented, for instance the term rotational viscosity is absolutely absurd, the sentence: "...  the rheological property of HCS-CRMA is generally superior to SILT-CRMA" is difficult to be interpreted.

Conclusion: Sorry to say but no evident progress from the first version can be detected.

Reviewer 4 Report

Most of my comments have been addressed well. 

Author Response

Response to Reviewer 4 Comments

Dear reviewer,

First and foremost, many thanks for your time and comments concerning our manuscript.

Those comments are all valuable and very helpful for helping us revise and improve the paper.  

We found that your response in the 'Comments and Suggestions for Authors' section was 'Most of my comments have been addressed well'. However, in the Open Review selection, it is 'I would not like to sign my review report'. Are you considering whether you clicked by mistake?. You may need to select 'I would like to sign my review report'.

Thank you very much again for your comments and suggestions. We appreciate for your warm work earnestly .

Best wishes,

Lu Huang

Round 3

Reviewer 3 Report

Sorry to say but the overwhelming majority of my comments was completely ignored.